# Language Models Need Sleep: Learning to Self Modify and Consolidate Memories

## Abstract

The past few decades have witnessed significant advances in designing machine learning algorithms–from early studies on task-specific shallow models to more general deep Large Language Models (LLMs). Despite showing promising results in tasks that requires instant prediction or in-context learning, existing models lack the ability to continually learn and effectively transfer their temporal in-context knowledge to their long-term parameters. Inspired by human learning process, we introduce a "*Sleep*" paradigm that allows the models to continually learn, transfer their short-term fragile memories into stable long-term knowledge, and self-modify themselves with "*Dreaming*" process. In more details, sleep consists of two main stages: (1) Memory Consolidation: a parameter expansion stage with a new Reinforcement Learning (RL)-based upward distillation process, called Knowledge Seeding, where the memories of a *smaller* model are distilled into a *larger* network to provide more capacity; (2) Dreaming, a self-improvement phase where the model uses Reinforcement Learning (RL) to generate a curriculum of synthetic data to rehearse new knowledge and refine existing capabilities without human supervision. Our experiments on long-context, continual learning, knowledge incorporation, and few-shot generalization tasks support the importance of the sleep stage and its contributions to improving the continual learning capability of the models.

## 1 Introduction

The development of Large Language Models (LLMs) marks a pivotal milestone in machine learning research: a paradigm shift from task-specific models to more general-purpose systems with various emergent capabilities (Brown et al., 2020; Schaeffer et al., 2023). Despite LLMs' remarkable capabilities in diverse sets of tasks (Wang et al., 2023; Nijkamp et al., 2023; Comanici et al., 2025), they are largely static after their initial deployment, meaning that they successfully perform tasks learned during pre- or post-training, but are unable to *continually acquire* new capabilities beyond their immediate context. This inherent static nature creates a crucial vulnerability: The model's knowledge and skills become progressively stale, operating with a fixed "knowledge cutoff" date beyond which it is unaware of new facts, events, and evolving information (Cheng et al., 2024).

Efforts to overcome this limitation have primarily focused on: (1) re-pretraining on an expanded dataset, which despite its effectiveness, is computationally expensive and impractical for frequent updates (Ibrahim et al., 2024); (2) using continual parameter updates, such as expensive Test Time Training (Sun et al., 2020; hongzhou yu et al., 2025), or other lightweight alternatives like fine-tuning or low-rank adaption (Hu et al., 2022; Akyürek et al., 2024a), which with iterative updates often results in Catastrophic Forgetting (CF) (Kemker et al., 2018; Shi et al., 2024)–a well-known phenomenon where the model's proficiency on original tasks degrades catastrophically as it learns new ones. This dilemma—between knowledge obsolescence on one hand and catastrophic forgetting as well as the prohibitive cost or destructive nature of updates on the other—underscores a critical, unresolved challenge: enabling LLMs to learn incrementally and efficiently throughout their lifecycle.

In recent years, In-Context Learning (ICL) (Brown et al., 2020) has gained attention as a highly efficient and successful form of continual learning (Akyürek et al., 2022; Dong et al., 2024; Akyürek et al., 2024b; Li et al., 2025). Initially, ICL was known as an emergent ability of LLMs that is

trained on large scale data, enabling them to adapt fast to the context and so perform zero- or few-shot tasks (Brown et al., 2020). Later, more studies revealed and formalized the role of ICL as a meta-learning process in which the model performs internal computations along the sequence to incorporate context knowledge to its output by keeping or compress it into a short-term memory (Author, s; Dherin et al., 2025). Despite the effectiveness/efficiency of ICL as a form of continual learning, it is limited to the context-window of sequence models, meaning that any new acquired knowledge will be removed from the model at the end of the session/context. This perspective raises a critical question:*How the model can effectively transfer the fragile short-term memories into more stable long-term knowledge*?

As an analogy, consider an impairment in the process of transferring the information from short-term to longer-term memories in humans, example of which is anterograde amnesia–a neurological condition where a person cannot form new memories after the onset of the disorder, while existing memories remain intact (Scoville & Milner, 1957). Such conditions can limit the person's knowledge to immediate present that fits in the short-term memory and long past, before the onset of the disorder, resulting in continuously experiencing the immediate present as if it were always new. One might notice a similar pattern in the memory processing of current LLMs. The knowledge of LLMs are limited in either: (1) the immediate context that fits into their context window (a.k.a. in-context learning), or (2) MLP and projection layers, storing long-past, before the onset of "end of pre-training." This similarity in pattern motivates us to ask, *What is the critical component in human learning process that consolidates memories?*

### THE ROLE OF SLEEP IN HUMAN LEARNING PROCESS

Sleep is not a passive state but a dynamic and highly structured period of brain activity essential for cognitive function (Rasch & Born, 2013; Goldstein & Walker, 2014). During sleep, the brain orchestrates complex processes fundamental to learning, neural plasticity, self-improvement, and memory consolidation (Wamsley & Stickgold, 2011; Rasch & Born, 2013; Goldstein & Walker, 2014). In humans, these processes are primarily governed by two critical and alternating stages of sleep: Rapid Eye Movement (REM) and Non-REM (NREM) sleep.

**Non-Rapid Eye Movement Sleep (Slow-Wave Sleep):** This stage, particularly its deepest phase known as slow-wave sleep, is characterized by synchronized, high-amplitude, low-frequency neural activity. Slow-wave sleep is associated with two primary functions crucial for learning: The first is synaptic homeostasis, a process that globally downscales synaptic strengths to counteract the net increase in connectivity from waking experiences, thereby maintaining metabolic balance and preventing neural saturation (Tononi & Cirelli, 2006).

The second core function is memory consolidation, the transformation of fragile, recent experiences into stable, long-term knowledge (Squire & Alvarez, 1995). This process is orchestrated through a sophisticated dialogue between the hippocampus and the neocortex (Squire et al., 2015). The hippocampus serves as a high-fidelity temporary storage system, capable of rapidly encoding specific daily experiences. In contrast, the neocortex is a vast, long-term repository better suited for the gradual learning of generalized rules and semantic knowledge from these experiences (Squire & Alvarez, 1995; Squire et al., 2015). During slow-wave sleep, the brain initiates a nightly dialogue between these structures that facilitates an intricate transfer of information. Notably, this transfer does not simply replay raw data; instead, it re-architects the knowledge acquired during waking hours, extracting abstractions and integrating them into a cohesive semantic network.

**Rapid Eye Movement Sleep:** Characterized by high-frequency, low-amplitude brain waves that resemble an awake state, REM sleep is most commonly associated with dreaming. Functionally, this stage is linked to the selective strengthening of newly formed synapses and the integration of new information with pre-existing emotional and semantic networks. Furthermore, it is hypothesized to play a role in simulating future scenarios to improve adaptive behavior.

In summary, the cyclical alternation between NREM and REM stages throughout the night is crucial. NREM sleep appears to consolidate and prune the day's experiences to build a more efficient knowledge base. Subsequently, REM sleep seems to operate on this refined base, exploring novel connections and strengthening salient neural pathways.

CONTRIBUTIONS

Inspired by the memory processing in humans, we introduce a "sleep" paradigm for LLMs, allowing them to consolidate their memories, and modify/improve themselves over time. Particularly, LLMs' sleep paradigm consists of two integrated phases:

1. **Memory Consolidation**: As discussed earlier, catastrophic forgetting (CF) is one of the main challenges to unlock the continual learning in LLMs. Contrary to recent efforts to mitigate catastrophic forgetting by developing more advanced encoding methods (Cheung et al., 2019; Fang et al., 2025), or sophisticated memory management (Irie et al., 2022; 2025; Author, s), we attribute CF as a direct consequence of limited capacity. To this end, we present a new method with gradual parameter growth over time that allows enough plasticity for new parameters, while ensuring the stability of old parameters. Particularly, this process is accompanied by knowledge seeding, an upward distillation process that distills the context of fast-updated and higher-frequency memory modules (such as ICL) into more stable, sparse, and vast parameters of feed-forward networks. To overcome the limitation of traditional distillation methods in distribution mismatch between output sequences seen during training and those generated during inference (Pomerleau, 1991; Ross & Bagnell, 2010), we present a variant of Generalized Knowledge Distillation (GKD) (Agarwal et al., 2024) that specifically motivates the student to *memorize* the knowledge abstractions learned by the teacher. Notably, the knowledge seeding step requires self-generated data , which can also be interpreted as a part of "Dreaming" stage.

2. **Self-Improvement via Dreaming**: While the previous first stage of sleep ensures transferring the knowledge abstraction to longer-term memories, this stage is responsible for the process of self-improvement. In particular, given the current state of the LLM, it generates a set of dreams–synthetically self-generated data to improve the performance with particular focus on the acquiring more proficiency on the recently added knowledge.

We evaluate the effectiveness of sleep paradigm on a set of challenging downstream tasks: (1) Factual Knowledge Incorporation; (2) Few-shot Learning; (3) Long-context Understanding; and (4) Continual Learning. The results support the effectiveness of Sleep paradigm as well as the importance of growing parameters with iterative knowledge distillation for continual learning.

## 2 PRELIMINARIES AND PROBLEM FORMULATION

In this section, we first discuss the notation we use throughout the paper and then review preliminaries background concepts that we build on. For the sake of clarity, we present a minimal discussion of related concepts in this section and provide a more in-depth review of backgrounds in Appendix A.

### 2.1 NOTATION

We use bold lowercase (resp. uppercase) letters for vectors (resp. matrices) and use subscript $t$ to refer to the state of the entities correspond to time $t$. Superscripts for parameters of a module (resp. hyperparameters) are used to determine the update frequency of the module (resp. distinguish different instances). Through the paper, we let $x \in \mathbb{R}^{L \times d_{\text{in}}}$ be the input, $\mathbf{K}$ be the keys, $\mathbf{V}$ be the values, $\mathbf{Q}$ be the query matrices in the sequence model, and $L$ denote the sequence length. When it is needed, we parameterize the language model $\text{LM}_\theta$ with $\theta = \{W_1^{(f_1)}, \ldots, W_{k_1}^{(f_1)}\} \cup \{W_1^{(f_2)}, \ldots, W_{k_2}^{(f_2)}\} \cup \ldots \{W_1^{(f_c)}, \ldots, W_{k_c}^{(f_c)}\}$, where parameter sets are sorted based on their weight update frequencies $f_1 \geq \ldots, \geq f_c$ (see Definition 1).

### 2.2 CONTINUUM MEMORY SYSTEM

Transformer architectures consist of two critical components: (1) Attention module that acts as an associative memory and conditions the output on the past tokens in the context, which also results in in-context learning ability; and (2) MLP or feedforward layers, which are fixed after the training phase and encodes the knowledge acquired over the pre-training. As discussed by Author (s), one can interpret such architectures as two-level memory systems, in which the attention's update span is the context length–meaning that at the end of the context, its corresponding parameters are updated

and the acquired knowledge is forgotten–and MLP's update span is non-existence–indicating no update after pre-training. From this perspective, these two components are two extreme sides of the frequency spectrum, where the attention (resp. MLP) has infinite (resp. zero) update frequency.

Building on the above intuition, Author (s) presented Continuum Memory System (CMS), where the architecture is a sequence model such as attention, followed by a chain of feedforward layers, each of which with its own update frequency. More specifically, the time for one step of update in the slowest module is considered as the unit of time, and so the update rate of other components are defined as:

**Definition 1 (Update Frequency)** *For any weight component of $W$, we define its frequency, denoted as $f_W$, as its number of updates per unit of time.*

To better understand this concept, we use a simple example of Fast-weight Programs (Schmidhuber, 1992), where the input is a sequence of length $L$. In this case, for each step of slow-weight (the unit of time), the fast-weight is updated $L$ times, resulting in update frequency of $L$.

Following this definition of frequency, which at high-level indicates how often the parameters of a module are updated over time, CMS is formalized as a chain of MLP blocks $\text{MLP}^{(f_1)}(\cdot), \ldots, \text{MLP}^{(f_k)}(\cdot)$, each of which associated with a chunk size of $C^{(\ell)} := \frac{\max_\ell C^{(\ell)}}{f_\ell}$ such that given input $x = \{x_1, \ldots, x_T\}$ the output of the chain is calculated as (we disregard normalizations for the sake of clarity):

$$y_t = \text{MLP}^{(f_k)}\big(\text{MLP}^{(f_{k-1})}(\cdots \text{MLP}^{(f_1)}(x_t))\big), \tag{1}$$

where the parameters of $\ell$-th MLP block, i.e., $\boldsymbol{\theta}^{(f_\ell)}$, are updated every $C^{(\ell)}$ steps:

$$\boldsymbol{\theta}_{i+1}^{(f_\ell)} = \boldsymbol{\theta}_i^{(f_\ell)} - \begin{cases} \sum_{t=i-C^{(\ell)}}^{i} \eta_t^{(\ell)} f(\boldsymbol{\theta}_t^{(f_\ell)}; x_t) & \text{if } i \equiv 0 \pmod{C^{(\ell)}}, \\ 0 & \text{otherwise.} \end{cases} \tag{2}$$

Here $\eta_t^{(\ell)}$ are learning rates corresponds to $\boldsymbol{\theta}^{(f_\ell)}$, and $f(\cdot)$ is the error component of an arbitrary optimizer (e.g., $\nabla \mathcal{L}(\boldsymbol{\theta}_t^{(f_\ell)}; x_t)$ in gradient descent). The conventional Transformer block (Vaswani et al., 2017) is a special instance of this formulation, where $k = 1$. Due to this generality of formulation, throughout the paper, we use c-MLPs as the default building blocks of the architectures. Also, for the sake of clarity and without loss of generality, we assume that $C^{(\ell)}$ is divisible by $C^{(\ell-1)}$. It is notable that Equation 2 provides an important interpretation: parameters $\boldsymbol{\theta}_t^{(f_\ell)}$ are responsible for compressing their own context into the their parameters and so they are a representative of abstract knowledge of their context (see Section A.1 for more details).

In summary, in this perspective, the sequence model (e.g., attention (Vaswani et al., 2017) or other memory modules or RNNs (Katharopoulos et al., 2020)) acts as the short-term memory of the model since their high-frequency update can push the old knowledge to be forgotten, making space for new memories. On the other hand, c-MLP blocks act as a spectrum of memory modules, in which earlier blocks (higher-frequency) are shorter-term memories, while later blocks (and ultimately the last one with close to zero frequency) are longer-term memories. While actively updating this memory system can enhance the resistance to CF, the CF can happen when the update period of all models matched at some point (Author, s). Therefore, it is crucial that before each update of a memory block, a mechanism consolidate the abstracted knowledge of that block to more stable parameters.

## 3 THE SLEEP PARADIGM

In this section, we present Sleep paradigm, in which contrary to the model's waking time (or active time), the model does not receive any external input data and concentrates its internal computations on self-improvement, consolidating the past memories, and abstracting knowledge. In particular, we divide sleep process into two key stages: (1) Memory consolidation; and (2) Dreaming for self-improvement.

### 3.1 MEMORY CONSOLIDATION: PARAMETER EXPANSION

As discussed earlier, in memory consolidation, we aim to transfer the short-term fragile memories into more vast and stable parameters. One of the important messages in CMS formulation is: the

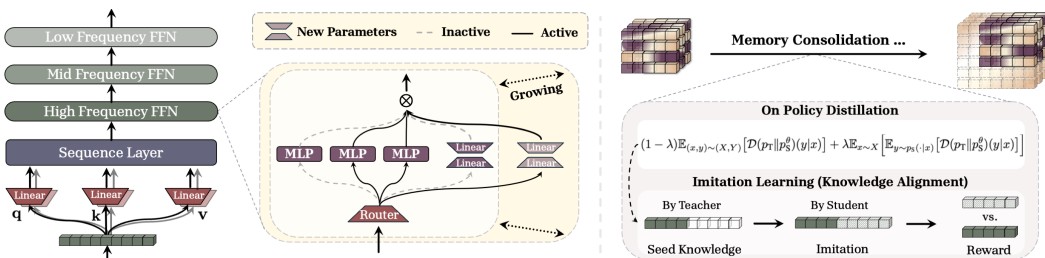

Figure 1: An overview of Memory Consolidation step. The model initially increases its own number of parameters to enhance its capacity (Section 3.1). Next, using our knowledge seeding, it transfers the knowledge abstractions from the higher-frequency memory to a lower one (Section 3.2).

fragility and/or stability of memories are relative. That is, for each memory block, other higher frequency memories are shorter-term and more fragile. Therefore, memory consolidation is not a simple two-step process, but an iterative operation that repeatedly transfers the knowledge stored in higher frequency memories into more stable lower-frequency parameters.

To avoid losing the knowledge of a faster updating block; an example of which is in-context learning (Brown et al., 2020), we need to perform memory consolidation step before updating its parameters. Therefore, given the list of chunk lengths $\{C^{(1)}, \ldots, C^{(k)}\}$, the sleep process and so memory consolidation happens only at $\{C^{(1)} \times b, \ldots, C^{(k)} \times b\}$ steps for all $b \in \mathbb{N}$. Based on the update frequency of MLP blocks, we might need to consolidate the memory of a memory module into its next memory block multiple times. For example, consider a memory with update frequency of 1K followed by a memory with update frequency of 10K: in this case, the faster memory is updated 10 times before the update of slower memory block, which means 10 memory consolidation steps of faster memory to slower memory before slower memory's own update. This multiple consolidation steps into the slower memory can be a critical bottleneck to unlock continual learning capabilities of LLMs, due to the Catastrophic Forgetting (CF). This phenomenon is an inherent cause of model's limited capacity (e.g., number of parameters), where parameters need to be overridden to incorporate the new knowledge. The foundation of human's brain solution to this challenge is neuroplasticity, the brain's inherent ability to modify its own function and shape new connections in response to experiences. Inspired by this, we present an efficient gradual parameter expansion in memory blocks that allows the model to shape new connections and so increases its own capacity.

Without loss of generality, we assume that the MLP blocks $\{\texttt{MLP}^{(f_\ell)}(\cdot)\}_{\ell=1}^k$ are sparse mixture of experts (MoEs) with a router $\mathcal{R}^{(f_\ell)}$: i.e., each $\texttt{MLP}^{(f_\ell)}(\cdot)$ includes a set of experts $\{W^{(f_\ell),1}, \cdots, W^{(f_\ell),\mathbf{s}_\ell}\}$, where $\mathbf{s}_\ell \geq 1$ is the current number of experts in the $\ell$-th block of the chain. Let $(\ell^* - 1)$ be the index of the memory (or MLP) that we aim to consolidate its knowledge to its immediate next more stable memory module with index $\ell^*$. To avoid the interference of transferred and previously stored knowledge in $\texttt{MLP}^{(f_{\ell^*})}(\cdot)$, we add a new low-rank expert to its set of parameters. That is, we add a low-rank MLP parametrized by $\{\mathbf{A}^{(f_{\ell^*}),\mathbf{s}_{\ell^*}+1}, \mathbf{B}^{(f_{\ell^*}),\mathbf{s}_{\ell^*}+1}\}$, where $\mathbf{A}^{(f_{\ell^*})} \in \mathbb{R}^{d \times d_{\text{low}}}$ and $\mathbf{B}^{(f_{\ell^*})} \in \mathbb{R}^{d_{\text{low}} \times d}$ ($d_{\text{low}} \ll d$), to the set of experts. These new parameters will be allocated for storing the new transferred knowledge from $\texttt{MLP}^{(f_{\ell^*-1})}(\cdot)$. Given this process, after each sleep time, the parameters of a subset of layers are growing.

## 3.2 MEMORY CONSOLIDATION: KNOWLEDGE SEEDING

In this step, we aim to transfer the knowledge of $\texttt{MLP}^{(f_{\ell^*-1})}(\cdot)$ with parameters $\boldsymbol{\theta}^{(f_{\ell^*-1})}$ into the expanded set of parameters in $\texttt{MLP}^{(f_{\ell^*})}(\cdot)$. We let $\texttt{LM}_{\boldsymbol{\theta}}$ be the state of the language model before parameter expansion and $\texttt{LM}_{\boldsymbol{\theta}_{\text{exp}}}$ be the state of language model after (i) parameter expansion, and (ii) updating of $\boldsymbol{\theta}^{(f_{\ell^*-1})}$ based on Equation 2. Note that since sleep and so memory consolidation is happening for $\texttt{MLP}^{(f_{\ell^*-1})}(\cdot)$, the number of past steps is divisible by $C^{\ell^*-1}$ and so this memory block is updated. We model the memory consolidation process as a distillation problem where we aim to transfer the knowledge stored in smaller state of the model $\texttt{LM}_{\boldsymbol{\theta}}$ to a larger variant of $\texttt{LM}_{\boldsymbol{\theta}_{\text{exp}}}$.

This distillation process has two critical challenges: (1) Contrary to conventional cases, student has more capacity and so more expressive power than the teacher. Therefore, training the student on the teacher generated dataset (e.g., Kim & Rush (2016)) can result in sub-optimal use of parameters in student model; (2) The model is in sleep stage and so the access to the external information/dataset is limited. Therefore, most popular methods like Hinton et al. (2015) are not applicable in this scenario. To overcome these challenges, we build upon Generalized Knowledge Distillation (GKD) (Agarwal et al., 2024), which allows a mixture of on-policy student generated data with a teacher-generated data, and present a novel distillation process based on imitation learning.

As discussed earlier, the memory consolidation step should not simply replay raw data; instead, it needs to explore and extract abstractions of knowledge acquired during active (waking) steps. To this end, knowledge seeding has two main steps: (1) A distillation process, in which student receives token-specific feedback from the teacher's logits on the self-generated sequences; and (2) An RL-based imitation learning method that forces the student to memorize the sampled outputs of teacher, aligning their sampling process while preserving the distilled knowledge.

We start with constructing a dataset $\mathcal{D}$ by sampling from the teacher model, i.e., $\text{LM}_{\boldsymbol{\theta}}$. Next, similar to GKD (Agarwal et al., 2024), we define the on policy distillation objective as:

$$\mathcal{L}(\boldsymbol{\theta}, \boldsymbol{\theta}_{\exp}) = (1-\lambda)\mathbb{E}_{(x,y)\sim\mathcal{D}}\big[\mathcal{F}(\text{LM}_{\boldsymbol{\theta}}\|\text{LM}_{\boldsymbol{\theta}_{\exp}})(y|x)\big] + \lambda\mathbb{E}_{x\sim\mathcal{D}}\Big[\mathbb{E}_{y\sim\text{LM}_{\boldsymbol{\theta}_{\exp}}(\cdot|x)}\big[\mathcal{F}(\text{LM}_{\boldsymbol{\theta}}\|\text{LM}_{\boldsymbol{\theta}_{\exp}})(y|x)\big]\Big],$$

where $\mathcal{F}(\text{LM}_{\boldsymbol{\theta}}, \text{LM}_{\boldsymbol{\theta}_{\exp}})(y|x)$ is a divergence between teacher (i.e., $\text{LM}_{\boldsymbol{\theta}}$) and student (i.e., $\text{LM}_{\boldsymbol{\theta}_{\exp}}$) output distributions, and $\lambda \in [0, 1]$ controls the the fraction of on-policy student-generated outputs. In this optimization process, we do not backpropagate through the sampling distribution of the student, which can help with training stability and also speed. Also, in this distillation process, we freeze all the parameters in the student model and only updates the expanded parameters. This ensures that the transferred knowledge does not interfere with the old knowledge, causing catastrophic forgetting.

**Learning to Imitate.** The above distillation process ensures that the student new parameters store the knowledge encoded in the lower-frequency memory. However, we observe that despite having access to the knowledge, the student model has not learned to use it and so weakly mimics the sampling and performance of the teacher. To this end, we further improve the above distillation process by incorporating RL to teach model how to imitate the teacher sampling. Given a set of teacher generated data (dreams), $\mathcal{D}_T = \{d^{(1)}, \ldots, d^{(n)}\}$, Learning to Imitate (LTI) process first randomly samples a prefix from each $d^{(i)}$ and then asks the student model to complete the continuation. Given the student responses $\hat{d}^{(i)}$ the assigned reward is defined as:

$$r(\hat{d}^{(i)}; d^{(i)}; \text{LM}_{\boldsymbol{\theta}_{\exp}}) = \gamma \times r_{\text{sem}}(\hat{d}^{(i)}; d^{(i)}; \text{LM}_{\boldsymbol{\theta}_{\exp}}) + (1-\gamma) \times r_{\text{abs}}(\hat{d}^{(i)}; d^{(i)}; \text{LM}_{\boldsymbol{\theta}_{\exp}}), \quad (3)$$

where $r_{\text{sem}}(\cdot; \cdot; \cdot)$ (resp. $r_{\text{abs}}(\cdot; \cdot; \cdot)$) assigns a reward based on the semantic similarity (resp. absolute token-level similarity). For semantic similarity, we use a reward model that is frozen and rewards the student with 1 (resp. 0), if the semantic of $\hat{d}^{(i)}$ and $d^{(i)}$ are the same (resp. otherwise). On the other hand, absolute reward is defined based on the Levenshtein distance of the two sequences (denoted by $z(\cdot, \cdot)$): i.e.,

$$r_{\text{abs}}(\hat{d}^{(i)}; d^{(i)}; \text{LM}_{\boldsymbol{\theta}_{\exp}}) = \begin{cases} 1 - \frac{z(\hat{d}^{(i)}, d^{(i)})}{\max\{|\hat{d}^{(i)}|, |d^{(i)}|\}} & \text{if } z(\hat{d}^{(i)}, d^{(i)}) \leq z_0, \\ 0 & \text{otherwise,} \end{cases} \quad (4)$$

where $z_0$ is a similarity threshold. By incorporating the above LTI process to on-policy distillation, the knowledge seeding (KS) objective is defined as:

$$\mathcal{L}_{\text{KS}}(\boldsymbol{\theta}, \boldsymbol{\theta}_{\exp}) = \mathbb{E}_{x\sim\mathcal{D}}\Big[(1-\alpha)E_{y\sim\text{LM}_{\boldsymbol{\theta}_{\exp}}(\cdot|x)}\big[r(y)\big] - \alpha\,\mathbb{E}_{y\sim\text{LM}_{\boldsymbol{\theta}_{\exp}}(\cdot|x)}\big[\mathcal{D}(\text{LM}_{\boldsymbol{\theta}}\|\text{LM}_{\boldsymbol{\theta}_{\exp}})(y|x)\big]\Big], \quad (5)$$

where $\alpha \in [0, 1]$ controls the strength of the distillation compared to the LTI objective. Based on this objective, we update the new expanded parameters of the model and consolidate the memory/knowledge of high frequency memory into lower-frequency memory blocks. Now that the memories in $\text{MLP}^{(f_{\ell^*}-1)}(\cdot)$ are consolidated in $\text{MLP}^{(f_{\ell^*})}(\cdot)$, we reset all the low-rank parameters that previously (in past sleep periods) have been added to $\text{MLP}^{(f_{\ell^*}-1)}(\cdot)$, making its capacity available for future. This step, can be interpreted as a similar procedure of synaptic pruning in human brain, in which brain prunes connections that are unnecessarily and/or redundant (Li et al., 2017) to enhance its efficiency and performance.

**Note on the Implementation.** Implementing the growing sparse modules can be extremely challenging if it requires a direct change in the dimensionality of tensors in the implementation. Alternatively, we can initially have those parameters in the model, but masked them in the forward and backward pass, before their initial activation in a sleep stage. Interestingly, it also aligns with our understanding of human brain, where brain has (large but) fixed capacity and new components are not added over time. Instead, new connections between brain regions can shape through our life time, unlocking the activation of new neurons and resulting in more plasticity to learn new knowledge (Kandell et al., 2021).

### 3.3 DREAMING: A SELF-MODIFYING PROCESS

The previous stage, which involved freezing higher-frequency parameters and distilling their knowledge to lower-frequency memories acts similar to slow-wave stage of sleep (NREM) in humans, which is responsible for memory consolidation. In REM stage, however, the brain is highly active (even on par with waking time) and aims to self-modify and strengthen newly formed synapses by dreaming. Inspired by this, we aim to design a dreaming process that learns how to generate dreams (synthetic data) that can help itself to improve over time.

In practice, any synthetic data generation process for self-improvement (e.g., Pang et al. (2024); Huang et al. (2025); Zweiger et al. (2025)) can be incorporated in this stage. A critical consideration, however, is the risk of iteratively applying self-improvement in continual learning setup, which might cause catastrophic forgetting (Zweiger et al., 2025). In our experimental evaluation, we show that how our two-step design of sleep as memory consolidation and then dreaming as self-modifying process is more robust to catastrophic forgetting. As a proof of concept, we build upon the work of Zweiger et al. (2025), SEAL; however, there are three challenges to incorporate it in our sleep paradigm: (1) Due to the cost of supervised fine-tuning (SFT) in SEAL's inner-loop, it is limited to small number of self-edits (dreams in our terminology). (2) Potential catastrophic forgetting as the cause of iterative self-improvement in sleep periods. (3) The sampling process only samples from the existing knowledge space of the model, while one of the key roles of dreaming is to explore novel synthesis of memories (Stickgold, 2005).

Given a sampled task $(C, \tau)$, where $C$ is the context containing information relevant to the task and $\tau(\cdot)$ is a measure to asses the performance in the downstream evaluation, our "dreaming" process starts with generating $m \geq 1$ dreams with having $C$ in context. In the sampling process, each router in MoE blocks additionally chooses a random expert and so incorporates random irrelevant knowledge to the dreaming, learning the underlying patterns that are hidden from model's sight. For this step, we let $\{\text{DREAM}^{(i)}\}_{i=1}^m \sim \text{LM}_{\boldsymbol{\theta}}(\cdot|C)$. Next, we reject some of the generated dreams and only keeps the samples with the most potential in improving the model's performance. To this end, we take inspiration from the literature on gradient-based data selection (Wang et al., 2024; Pan et al., 2024): for each dream, $\text{DREAM}^{(i)}$, we assign an importance score $\boldsymbol{\omega}^{(i)}$ and select Top-$k$ dreams with highest importance score along with $b$ random samples to maintain diversity. Given language modeling objective $\mathcal{L}_{SFT}(\cdot)$, we define importance score of $\text{DREAM}^{(i)}$, denoted as $g_{\text{DR}}^{(i)}$, as the gradient of the objective:

$$g_{\text{DR}}^{(i)} = \nabla_{\boldsymbol{\theta}} \mathcal{L}_{SFT}(\text{DREAM}^{(i)}, \boldsymbol{\theta}). \tag{6}$$

We let D be the set of all selected dreams by the above process. For each $\text{DREAM}^{(i)} \in \text{D}$ we consider an isolated instance of the model and updates its parameters via supervised finetuning (with LoRA (Hu et al., 2022)): i.e., $\boldsymbol{\theta}'^{(i)} \leftarrow \text{SFT}\left(\boldsymbol{\theta}^{(i)}, \text{DREAM}^{(i)}\right)$. Given the new fine-tuned model, following SEAL (Zweiger et al., 2025), we reward the generation of $\text{DREAM}^{(i)}$ based on $\text{LM}_{\boldsymbol{\theta}'^{(i)}}$'s performance improvement over $\text{LM}_{\boldsymbol{\theta}^{(i)}}$:

$$r\left(\text{DREAM}^{(i)}, \tau(\cdot), \text{LM}_{\boldsymbol{\theta}^{(i)}}\right) = \begin{cases} 1 & \text{If, DREAM}^{(i)} \text{ improves LM}_{\boldsymbol{\theta}^{(i)}}\text{'s performance,} \\ 0 & \text{Otherwise.} \end{cases} \tag{7}$$

We follow SEAL and use ReST$^{EM}$ algorithm (Singh et al., 2024) to optimize the above process.

Table 1: Memory consolidation enhances the model long-context understanding.

| Method | ICL | Cartridges | Duo Attention | Sleep |
|---|---|---|---|---|
| MTOB | 34.7 | 35.1 | 35.6 | **40.3** |
| LongHealth | 53 | 54 | 49 | **59** |
| QASPER ($\downarrow$) | 1.3 | 1.2 | 1.5 | **1.1** |

## 4 EMPIRICAL RESULTS

### 4.1 EXTENDING THE EFFECTIVE CONTEXT WINDOW

In this section, we evaluate the effect of sleep on the ability of model to extrapolate to longer contexts than its context window size.

**MTOB, LongHealth, and QASPER Benchmarks.** In the first experiment we focus on the MTOB benchmark (Tanzer et al., 2023), which the model needs to translate from Kalamang, a low-resource language, into English. The data comprises diverse $(q, r)$ pairs anchored to a single lengthy source document. Also, as the architectural backbone of our LLM, we use LLAMA-8B. Since LLAMA-8B is a Transformer-based architecture, the number of memory blocks are two: attention block, which has the context length of 128K, and MLP layers, which we assign $5 \times 128K$ as their chunk size. Following previous studies we use character n-gram F-score (chrF) as the metric of this task (Tanzer et al., 2023; Eyuboglu et al., 2025). We use in-context learning (ICL), Cartridges (Eyuboglu et al., 2025), duo attention (Xiao et al., 2025), as the baselines. The results are reported in Table 1. The sleep paradigm makes the model more powerful and can outperform all the baselines, including ICL and Cartridges.

We further perform experiments on LongHelath and QASPER benchmarks. Again, we follow the setup in Eyuboglu et al. (2025) and use accuracy (resp. perplexity) as the metric for LongHelath (resp. QASPER) benchmark. The results are reported in Table 1. Sleep significantly outperforms in-context learning and compression methods. We attribute this performance to the memory consolidation process and its parameter growth step that enhances the capacity.

**BABILong Benchmark.** To evaluate the effectiveness of Sleep on the effective context length of the model, we further evaluate our LLAMA-8B-based model's performance on BABILong benchmark (Kuratov et al., 2024). In this experiment, we follow Kuratov et al. (2024) and use the original setup in the benchmark without fine-tuning the model. The results show that while our model shows competitive and on par performance with ultra-large models until 1M context length, it maintains its performance and achieve +90% accuracy in 10M context length. These results highlight the importance of iterative memory consolidation as the sequence model (e.g., attention) can fail to fully incorporate the knowledge in a long context.

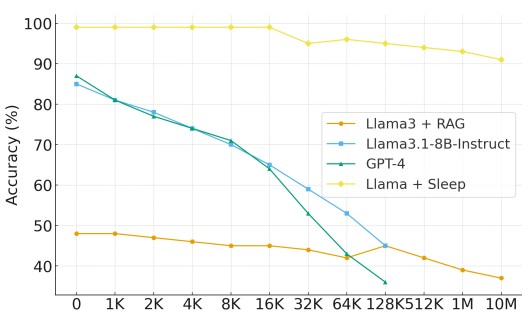

Figure 2: The performance of our model and baselines on BABILong benchmark (Kuratov et al., 2024).

Table 2: Knowledge Incorporation Performance across Passage Settings

| Method | Single Passage (n = 1) | Continued Pretraining (n = 200) |
|---|---|---|
| Base model | 31.9 | 31.9 |
| Fine-tuned Model with No Dreaming | 33.4 | 32.0 |
| SEAL | 46.7 | 43.2 |
| Sleep (Transformer) | 48.1 | 44.3 |
| Sleep (Transformer + four-level) | 48.9 | 46.2 |

## 4.2 KNOWLEDGE INCORPORATION

One of the important questions about the effectiveness of sleep paradigm is whether the model can incorporate new factual knowledge. With an effective memory consolidation process, we expect the model to be able to answer questions about the incorporated facts. We follow the experimental setup of Zweiger et al. (2025), including the choice of models and parameters. We evaluate our model on integrating new factual information from SQUAD dataset (Rajpurkar et al., 2016). As baselines, we use (i) a base model, which is the variant without any improvement or having access to the passage; (ii) a fine-tuned model with no dreaming, (iii) SEAL model with RL and self-adaption; (iv) our Transformer-based architecture with two level memory system; and (v) our Transformer-based architecture with four-level memory system.

Table 2 summarizes mean no-context SQuAD accuracy for both the single-passage ($n = 1$) and continued pretraining (CPT, $n = 200$) settings. Our sleep process achieves the best results among other settings and state-of-the-art methods like SEAL. We attributes this results to: (1) memory consolidation steps that let the model store its knowledge more effectively; (2) our improvements on top of the SEAL that we discussed in Section 3.3. In the CPT regime, the model is exposed to $n = 200$ passages during a single continued pretraining run and is evaluated on the full set of 974 associated questions. For each passage, we sample five dreams and combine them into an aggregated synthetic dataset for training. Again, our sleep process achieves the best performance.

## 4.3 FEW-SHOT LEARNING

We follow the few-shot ARC experimental protocol from prior work (Akyürek et al., 2024a; Zweiger et al., 2025) and adapting it to our SLEEP paradigm. As the backbone we use `Llama-3.2-1B`. Following common practice, we filter subset of data to avoid tasks that remain unsolvable under standard configurations, yielding 11 tasks for training and 8 held-out tasks for evaluation.

During each *Sleep* cycle, the model first consolidate its previous memories, and then *dreams* by generating synthetic experiences from the few-shot demos. For each task, we sample 60 dreams and reject 45 of them. At test time, for each unseen task, the model generates 5 dreams and applies them independently before predicting the held-out output. We report the fraction of dreams that yield a correct answer. As the baselines, we follow Zweiger et al. (2025) and use: (i) ICL (In-Context Learning); (ii) TTT + synthetic updates (no dreaming); and (iii) SEAL (Zweiger et al., 2025).

In this setting, SLEEP achieves a 80% success rate, higher then the other methods.

Table 3: Few-shot Abstract Reasoning

| Method | Success Rate (%) |
|--------|------------------|
| ICL    | 0                |
| TTT    | 10               |
| SEAL   | 72.5             |
| Sleep  | 80               |

## 5 CONCLUSION

In this work, we introduced the SLEEP paradigm for Large Language Models, which alternates between a waking phase and an offline *sleep* phase comprising (i) *knowledge seeding*—an upward distillation that transfers short-term, in-context knowledge into lower-frequency, long-term parameters—and (ii) *dreaming*—selective, self-generated training that improves capabilities while controlling interference. Across long-context understanding, knowledge incorporation, few-shot reasoning, and continual learning, SLEEP yields consistent gains over ICL, compression-based baselines, and self-adapting methods, while reducing memory pressure. Remaining challenges include amortizing the cost of sleep cycles and further automating sampling/pruning policies; future work will explore online scheduling, alignment-aware dreaming, and multi-agent sleep. By structuring learning as alternating consolidation and self-improvement, SLEEP moves LLMs toward stable, lifelong learning.

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

# A    PRELIMINARIES AND BACKGROUND

In this section, we provide a more comprehensive discussion of preliminaries and background concepts.

**Attention.** Attention is the primary building block of Transformers that acts as their short-term associative memory (Bietti et al., 2023; Sun et al., 2024; Behrouz et al., 2025). Given input $x \in \mathbb{R}^{L \times d_{\text{in}}}$, causal attention computes output $\mathbf{y} \in \mathbb{R}^{L \times d_{\text{in}}}$ over input dependent key, value, and query matrices $\mathbf{Q} = x\mathbf{W_Q}, \mathbf{K} = x\mathbf{W_K}$, and $\mathbf{V} = x\mathbf{W_V}$ as:

$$\mathbf{y}_i = \sum_{t=1}^{i} \frac{\exp\left(\mathbf{q}_i^\top \mathbf{k}_t\right)\mathbf{v}_t}{\sum_{\ell=1}^{i}\exp\left(\mathbf{q}_i^\top \mathbf{k}_\ell\right)} = \frac{1}{Z_i}\sum_{t=1}^{i}\exp\left(\mathbf{q}_i^\top \mathbf{k}_t\right)\mathbf{v}_t, \tag{8}$$

where $\mathbf{W_Q}, \mathbf{W_K}$, and $\mathbf{W_V} \in \mathbb{R}^{d_{\text{in}} \times d_{\text{in}}}$ are learnable parameters, and $Z_i = \sum_{\ell=1}^{i}\exp\left(\mathbf{q}_i^\top \mathbf{k}_\ell/\sqrt{d_{\text{in}}}\right)$ is the normalization term. Despite Transformers' simple parallelizable training, their generation process and long-context scaling are significant drawbacks, as attention requires at least $L \times d$ operations per token to calculate the output.

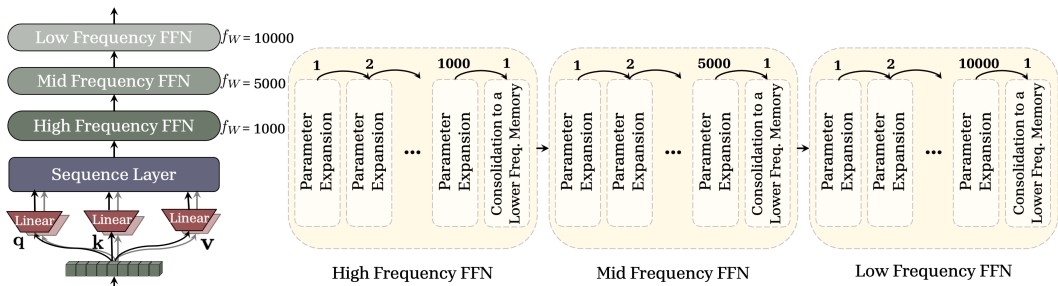

Figure 3: Multi-frequency memory hierarchy. Updates enter the High-Frequency FFN via repeated Parameter Expansion; when the window $f_W$ expires, knowledge is Consolidated to the Mid- and then Low-Frequency FFNs (1k→5k→10k).

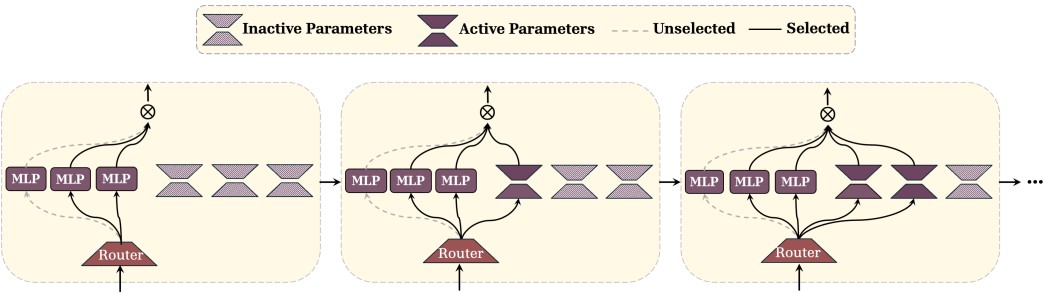

Figure 4: Memory consolidation by routed expert updates. Across Sleep cycles (left→right), a router selects and updates a small set of experts (solid), leaving others inactive (hatched), expanding capacity while limiting interference.

## A.1    NESTED LEARNING

Additional information about Nested Learning and C-MLP architecture can be found in this anonymous draft: This Link

