# OpenReview forum: "Language Models Need Sleep: Learning to Self Modify and Consolidate Memories"
_ICLR.cc/2026/Conference — Submitted to ICLR 2026_

### Official Review · Reviewer_4wzC · 2025-10-17

**Soundness:** 2
**Presentation:** 1
**Contribution:** 2
**Rating:** 2
**Confidence:** 3

**Summary:**

This paper seeks to expand the knowledge in a Transformer model after it has been trained without incurring in catastrophic forgetting. For this it proposes to adopt a MoE architecture in the MLP component of the Transformer, which is incrementally expanded training over time. Furthermore, multiple copies of these components are stacked, with some of them being updated more frequently than others. Finally it proposes to use student-and-teacher distillation paradigm supplemented with a RL loss to consolidate knowledge from high-frequency components to low-frequency ones.

**Strengths:**

- S1) The problem of continual learning without catastrophic forgetting is long-standing in ML with many existing proposals but not yet a clear universally accepted solution. Thus, the paper attacks a problem that is worth of study.
- S2) While some of the ideas have been explored before (e.g. fast adapting weights, growing models, experience replay), maybe this exact combination was not explored.
- S3) The method performance is better than the baselines as claimed by the paper.

**Weaknesses:**

- W1) This paper was extremely hard for me to read, even though I am overall familiar with the continual learning literature and the Transformers architecture. I try to identify what I found unclear through the questions section so the authors can know where to improve.
- W2) The model is somewhat complex involving a number of components such as a) differentially-updated-weights, b) distillation, c) Imitation learning via RL. However, the contribution of each of these parts is not studied via ablation experiments. In particular the concept of "self-improving via dreaming" sounds as an over-claim in the face of the reported results.
- W3) There is close to no discussion of the vast literature on continual learning. Instead, there is a though discussion of neuroscience literature on dreaming (which as a matter of fact I suspect was AI-generated because of the markedly different tone that it has with respect to the rest of the paper) that is not relevant as the relation with dreaming is only a metaphore. If the authors should want to establish a more clear connection with human dreaming, then they should at least show that their model explains at some level human behavioral data.
- W4) The paper compares to methods for long-context processing (Cartdiges and Duo Attention) and a self-adaptation model (SEAL) that doesn't deal with catastrophic forgetting. Instead, there is close to no comparison to the vast field of continual learning.
- W5) There is no report on hyper-parameter usage, implementation details, nor existing code, making the reproducibility of the results questionable at best.
- W6) The presentation overall is a bit confusing in its structure which reads as a sequence of loosely connected sections. The reference of a paper by anonymous authors (presumably the authors of the submission) which is nowhere to be found makes it all the more complicated to read because I've tried to understand more the model that they are expanding upon here by looking at this publication to no avail.

**Questions:**

- Q1) "we attribute CF as a direct consequence of limited capacity" -> This is a strong and interesting claim! Have you validated it? E.g. If you train Gemma 27B on some arbitrary sentence set won't it suffer from CF?
- Q2) Could you say more about "self-improving" via dreaming? What does an RL objective give you that max likelihood doesn't?
- Q3) What is the usefulness and distinctive quality of the "dreaming" metaphor? There already exists the concept of experience replay. How dreaming is different here? Note also that this metaphor was already quite prominently used for something else: https://www.science.org/doi/10.1126/science.7761831
- Q4) What is the relation between "forgetting" at the level of the activations resulting from an input context that changes and "forgetting" due to the the update of the weight parameters? I've felt that these two concepts are confused throughout the paper and I would like to see it more clearly defined.
- Q5) What is the unit of time of the update frequency?
- Q6) I'm quite confused about the concept of the "chunk size". First, the definition of $C^{(l)}$ probably has some typo because it is defining an endless recursion. Furthermore, what is the notion of a "chunk" here? I understand a chunk as a sequence of tokens of a given size. However, it seems to be speaking only about the the period in number of tokens every which a component is updated. Consider, for example, a model with context length of 10, and two FF layers, the fast one with update frequency 1 and the slow one with update frequency 2. Then, as I understand it, the fast one will be trained on two sequences of length 10, whereas the slow one will be trained on a single sequence of length 10 (at least initially until more information comes through the student-teacher paradigm). Is there anything wrong in how I'm understanding it?
- Q7) L192: why there is no loss in generality?
- Q8) What is the meaning of "l" in your notation? MLP layer? Overall I find that time across sequence processing (measured in number of tokens), in single token processing (measured in number of layers), and in training (measured in number of gradient steps) are not very clearly used.
- Q9) L273: Why the access to external information is limited?
- Q10) Eq 6: this is a vector, not a score. How do you compute the score?
- Q11) Could you explain more why your approach helps with long-document processing?
- Q12) Could you give more details on the benchmarks you use, particularly on long document question answering?
- Q13) More of a comment than a question -- You should check out McClelland's Complementary Learning Systems theory and probably discuss connections to it if relevant: https://pubmed.ncbi.nlm.nih.gov/7624455/

Style typos and grammar:
- Abstract: "tasks that requires" -> "tasks that require"
- L234: "," instead of ";"
- The use and abuse of the x (resp. y) construction makes some parts really hard to read.
- Table 1: add units.

**Details Of Ethics Concerns:**

Just using this field to mention that the part that I believe was AI-generated is the one corresponding to the literature review on dreams, which is not very important as it should probably be removed.

---

### Official Review · Reviewer_T73n · 2025-10-29

**Soundness:** 1
**Presentation:** 1
**Contribution:** 1
**Rating:** 0
**Confidence:** 3

**Summary:**

This paper introduces a new “Sleep” method for continual learning, consisting of “memory consolidation” and “dreaming.” The method specifically applies to an architecture where self-attention layers are followed by MoE-MLP layers whose experts/MLPs are updated with different frequencies. The memory consolidation step involves generating synthetic data sampled from a language model before “parameter expansion” and using it to train a parameter-expanded version. This is reinforced by RL on the semantic similarity of sequences generated by the two models. Dreaming involves generating sequences and identifying the ones that improve the models’ performance on SFT (and then doing RL on this reward signal). Experiments are presented on MTOB/LongHealth/QASPER where Sleep beats ICL, Cartridges and DuoAttention; on SQuAD to verify retention of knowledge, and on ARC for few-shot learning.

**Strengths:**

1. The method improves upon DuoAttention in long-context evaluation.
2. The idea of having fast and slow updates to a model is interesting, although it seems to have been attributed to an anonymized prior work claimed to be by the same authors (more below).

**Weaknesses:**

1. **Anthropomorphization**. This paper claims that their method is like the REM and non-REM phases of sleep in the human sleep cycle. However, that is just an analogy for visualization; I do not see a technical reason, concrete experiments, or ablation studies that suggest any link.

2. **Reference to missing material**. The paper repeatedly refers to an anonymous paper on this new architecture with multiple MLPs per attention layer (which are updated with different frequencies, but also see next point). However, there is no link to an anonymous version, just to a paper by “Authors.” Without this background, I find it extremely difficult to understand what the paper is trying to describe.

3. **Time axis is ambiguous**. The paper mentions that the MLPs are updated with different frequencies of time. I am confused here: is time the axis of training steps, or the axis of the token index? The experiment on long-context benchmarks suggests that it is token index, but the writing suggests that it is the training steps. Also, if it is training steps, I am not sure it makes much sense: there is no link between successive examples seen in training, so I don’t get why we would want to learn concepts at different frequencies.

4. **Several aspects of the method are not described**. If I am understanding correctly,

    a. **Architecture**. We have an MoE-MLP layer where each expert is updated with a different frequency.

    b. **Memory consolidation**. The authors write “we model the memory consolidation process as a distillation problem where we aim to transfer the knowledge stored in smaller state of model LM_theta to larger variant LM_{theta_exp}. What does this mean? Do we create a fresh copy of the MLP with more experts (the figure suggests this is true, but there is no textual description)? More dimensions per expert? Or just a larger model?

    c. We then perform distillation from the older to the newer model. The authors call this “consolidation,” but I do not see why one would want to just train a new layer based on an already existing layer (also see comment on ablations later). Not to mention that these sequences are sampled from the older network, which is both costly and will yield poor sequences (especially early on).

    d. And then, we perform RL on the semantic similarity between the generations of the two models. Where do the prompts here come from? What is the RL objective? How expensive is the RL process? No details are given for an already complicated process.

    e. **Dreaming**: If I understand dreaming, this involves (i) generating rollouts to a prompt (ii) *actually training on each of the rollouts* (iii) assigning a reward based on whether the trained model actually performs well on some evaluation on benchmarks (iv) doing RL based on this reward.  But here (1) this sounds extremely expensive, how did you manage to train it? (2) what is the evaluation on, for the reward? (3) what are the details/hyperparameters/objective of the RL? (4) where do the prompts come from? Again, no details are given.

5. **No details or ablations are given for any of the components**. As mentioned above, there are many, very complicated parts of method. However, no details of the data, hyperpameters, computational requirements, etc. are provided. Also, none of the design choices are ablated; this is important here since all of the components are complex and computationally expensive. Of these, at least one stands out to me: the dreaming phase involves training on each of the rollouts -> evaluating -> using that as a reward. Since results are reported on 8B Llama models, this represents an absurdly high cost for training that would be *infeasible even with hundreds of GPUs*.

6. **The evaluation section is scattered**.  I am not sure why the method should be evaluated on long-context benchmarks. Even if it were, benchmarks like QASPER are long outdated—there should at least be evaluation on recent works like LongBench [1,2], Needle-In-A-Haystack [3], or HELMET [4]. The evaluation on SQuAD is really odd. First, SQuAD is a question-answering dataset based on Wikipedia articles, so why would you not train on Wikipedia instead of SQuAD? Also, this benchmark is very old, and since its from Wikipedia, virtually all recent models should obtain near-perfect accuracy without any training. Yet, the plot suggests that the initial model only gets 32% accuracy—what model is this? (also see below)

7. **The evaluation has several inconsistencies and contradictions**. Several parts of the evaluation stick out, such as:

    a. The model name for the long-context evaluation is inconsistent. The text says they use “Llama-8B,” but the figure has one baseline as “Llama3 + RAG,” and another as “Llama-3.1-8B-Instruct.” Which one of LLama(-1), Llama-3, Llama-3.1 (base/instruct) did you use?

    b. The evaluation on BabiLong does not make much sense either. Llama-3 and 3.1 have a maximum context length of 8K/128K tokens; yet the text evaluates on sequences up to 10M tokens, where the model would completely break down! (The original BabiLong paper stops evaluation for Llama-3.1-8B-Instruct at 128k tokens - where they report a score of 39, far below 45 as per the plot of this paper). Also: inference on a single GPU not work beyond 128k tokens even with the most extreme optimizations (and hence 10M would need special frameworks and over 64 GPUs, needing description), but the results even suggest that sleep, a non-RAG method, not just works but excels at a 10M context length. This result is extremely noteworthy and confusing without detailed analysis.

   c. In the SQuAD results, the results are reported with “a base model.” Which one? Given the very poor results of the base model, it is difficult to believe that it is one of the modern LLMs with >= 1B parameters from the past two years.

    d. Furthermore, Llama-3.1 is **not a Mixture-of-Experts model** - how was it managed to train the specified architecture with several MoE-MLP layers with this base model?

Especially based on points 3 (missing and where present, highly implausible details of the method) and 7 (highly implausible details of the evaluation), I believe that the claims of the paper require closer scrutiny—at least in the form of a much more detailed and clear explanation of the method and evaluation, and the experimental details.


[1] “LongBench: A Bilingual, Multitask Benchmark for Long Context Understanding,” Bai et. al., ACL 2024

[2] “LongBench v2: Towards Deeper Understanding and Reasoning on Realistic Long-context Multitasks,” Bai et. al., arXiv 2025

[3] “Needle in the Haystack for Memory-Based Large Language Models,” Nelson et. al., arXiv 2024.

[4] “HELMET: How to Evaluate Long-Context Language Models Effectively and Thoroughly,” Yen et. al., ICLR 2025.

**Questions:**

1. Can you clarify whether the time axis is along the training steps or the token index?
2. Can you provide more details of the RL step in both phases, but especially the dreaming phase?
3. Can you elaborate on how a model trained with sleep can function at 10M tokens if there is no RAG involved? Also, can you elaborate on what hardware optimizations were needed to enable this?

---

### Official Review · Reviewer_ptx1 · 2025-10-31

**Soundness:** 2
**Presentation:** 2
**Contribution:** 2
**Rating:** 4
**Confidence:** 4

**Summary:**

The paper proposes a “sleep-inspired” paradigm motivated by human sleep mechanisms, aiming to address two core challenges faced by LLMs after deployment: knowledge obsolescence and catastrophic forgetting. The framework consists of two key phases—"Memory Consolidation" and "Dreaming"—which together facilitate continual adaptation. Experimental results on several benchmarks demonstrate that the proposed approach  enhances the model’s continual learning capability.

**Strengths:**

* This paper is well-motivated, and overall method design is interesting.
* The experiments cover tasks such as context window extension, knowledge integration, and few-shot learning

**Weaknesses:**

- The proposed paradigm has high design complexity, involving multiple stages of distillation, reinforcement learning, and supervised fine-tuning. The overall computational cost is likely substantial, yet no quantitative analysis of training or inference overhead is provided.

- Reproducibility and experimental details
    - Several key hyperparameters are missing. The source and independence of the semantic reward model used during the “dreaming” phase are not clearly described.

    - In the BABILong context extension experiments, the paper claims “without fine-tuning the model,” yet the overall framework includes multiple parameter updates.

    - The baselines in the long-context experiments rely on inference-time techniques, without matching training budgets, making the comparison potentially unfair.

    - In the knowledge incorporation experiments, the SEAL result for n = 200 differs significantly from that reported in the original paper.

    - Different model sizes are used across experiments; consistent model sizes or size-wise analyses would improve fairness and clarity.

- Experimental completeness: The paper lacks ablation studies to verify the contribution of each stage in the framework. Although the approach claims to mitigate catastrophic forgetting, most experiments focus on incremental knowledge acquisition, without evaluating the retention of old knowledge.

**Questions:**

I am not fully convinced that introducing the concept of human “sleep” mechanisms is necessary. While it provides an intuitive metaphor, it does not appear to strengthen the technical soundness of the method. The connection between biological sleep processes and LLM continual learning remains weak, and the analogy may add unnecessary conceptual complexity, making the paper harder to interpret.

---

### Official Review · Reviewer_J1XC · 2025-11-01

**Soundness:** 2
**Presentation:** 2
**Contribution:** 3
**Rating:** 4
**Confidence:** 3

**Summary:**

The paper introduces Sleep, a biologically inspired framework that lets language models "rest, dream, and grow".
During sleep, the model pauses normal training to consolidate recent knowledge into slower, deeper parameters through self-distillation and lightweight parameter expansion; Dream by generating synthetic experiences, replaying and refining what matters most.

This cycle aims to reduce forgetting, improve continual learning, and echo the way brains reorganize memories. Experiments suggest better long-context reasoning, factual recall, and few-shot generalization compared to existing self-improvement baselines.

**Strengths:**

- The paper brings together ideas from neuroscience and machine learning in a natural way.
- Its originality lies in treating a language model as a living system that learns, rests, and dreams, a poetic yet rigorous framing of continual learning.
- The methodology is internally consistent, showing steady improvements across tasks that matter.

**Weaknesses:**

The paper’s ideas are elegant, but its empirical grounding and exposition need more depth.

- Ablations: The framework weaves together expansion, distillation, and dreaming, yet their individual roles remain unclear. Focused ablations would reveal what truly drives improvement.

- Stability: The long-term effects of repeated dreaming, possible drift or amplified hallucination, are not examined. Assessing this would strengthen the work’s reliability.

- Context: The related-work section overlooks nearby directions such as “Improving Language Plasticity via Pretraining with Active Forgetting” (NeurIPS 2023) and parameter-efficient continual learning. The active forgetting view, periodic resetting rather than consolidation, offers a natural counterpoint and would help situate the “sleep” mechanism within the broader landscape of biologically inspired adaptation.

With deeper empirical support and a more connected discussion of prior work, the paper’s quiet but original vision would resonate more clearly.

**Questions:**

1. Which elements of the Sleep framework (expansion, distillation, dreaming) drive the main improvements?

2. How does performance evolve over many sleep cycles? Have you observed drift or hallucination buildup, and what safeguards help maintain stability?

3. How does your approach compare or interact with “Improving Language Plasticity via Pretraining with Active Forgetting” (NeurIPS 2023) and parameter-efficient continual learning? Could these methods complement the proposed consolidation process?

---

### Meta-Review · Area_Chair_AE18 · 2026-01-01

**Summary:**

Across reviews there is a strong and consistent concern that the paper lacks sufficient technical clarity, empirical rigor, and reproducibility. Most critically, Reviewer T73n raises severe red flags regarding missing method details, internal inconsistencies, computational implausibility of the reported experiments, and contradictions between the claimed architecture and evaluated models. These concerns go beyond presentation quality and call into question the validity of the results themselves. Reviewer 4wzC echoes many of these issues, emphasizing the lack of ablations, poor positioning within the continual learning literature, and confusing exposition. Overall, while the conceptual framing is novel, the current submission does not meet the bar for technical soundness or credibility expected at ICLR.

**Reviewer Concerns:**

No rebuttal was submitted.

**Reviewer Scores:**

No rebuttal was submitted.

---

### Decision · Program_Chairs · 2026-01-26

Reject